# A Weighted Linearization Method for Highly RF-PA Nonlinear Behavior Based on the Compression Region Identification

Jose Alejandro Galaviz-Aguilar [1,*], Cesar Vargas-Rosales [1], José Ricardo Cárdenas-Valdez [2], Yasmany Martínez-Reyes [2], Everardo Inzunza-González [3], Yuma Sandoval-Ibarra [2] and José Cruz Núñez-Pérez [4]

1 School of Engineering and Sciences, Tecnologico de Monterrey, Monterrey 64849, Mexico; cvargas@tec.mx
2 ITT, Department of Electrical and Electronics Engineering, Tijuana Institute of Technology, Tijuana 22435, Mexico; jose.cardenas@tectijuana.edu.mx (J.R.C.-V.); yasmany.martinez18@tectijuana.edu.mx (Y.M.-R.); jumasaniba@gmail.com (Y.S.-I.)
3 Engineering, Architecture and Design Faculty, UABC, Ensenada 22860, Mexico; einzunza@uabc.edu.mx
4 Instituto Politecnico Nacional, IPN-CITEDI, Av. Instituto Politecnico Nacional No. 1310, Colonia Nueva Tijuana, Tijuana 22435, Mexico; jnunez@ipn.mx
* Correspondence: galaviz@tec.mx

**Abstract:** In this paper, we present an adaptive modeling and linearization algorithm using the weighted memory polynomial model (W-MPM) implemented in a chain involving the indirect learning approach (ILA) as a linearization technique. The main aim of this paper is to offer an alternative to correcting the undesirable effect of spectral regrowth based on modeling and linearization stages, where the 1-dB compression point (P1dB) of a nonlinear device caused by memory effects within a short time is considered. The obtained accuracy is tested for a highly nonlinear behavior power amplifier (PA) properly measured using a field-programmable gate array (FPGA) system. The adaptive modeling stage shows, for the two PAs under test, performances with accuracies of −32.72 dB normalized mean square error (NMSE) using the memory polynomial model (MPM) compared with −38.03 dB NMSE using the W-MPM for the (i) 10 W gallium nitride (GaN) high-electron-mobility transistor (HEMT) radio frequency power amplifier (RF-PA) and of −44.34 dB NMSE based on the MPM and −44.90 dB NMSE using the W-MPM for (ii) a ZHL-42W+ at 2000 MHz. The modeling stage and algorithm are suitably implemented in an FPGA testbed. Furthermore, the methodology for measuring the RF-PA under test is discussed. The whole algorithm is able to adapt both stages due to the flexibility of the W-MPM model. The results prove that the W-MPM requires less coefficients compared with a static model. The error vector magnitude (EVM) is estimated for both the static and adaptive schemes, obtaining a considerable reduction in the transmitter chain. The development of an adaptive stage such as the W-MPM is ideal for digital predistortion (DPD) systems where the devices under test vary their electrical characteristics due to use or aging degradation.

**Keywords:** FPGA; memory effects; nonlinear modeling; power amplifiers; W-MPM

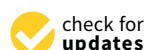



## 1. Introduction

The choice of the compression region for operation near saturation, thus avoiding the back-off driving of a power amplifier (PA), poses a transcendental challenge in terms of the search for a tradeoff between efficiency and linearity. This fact combined with addressing design issues and solutions in PAs adapted to the 5G mobile requirements, together with using new digital modulations for mobile applications and the high demand for applications, services, and mobile devices, becomes even more critical. Thus, to meet user capacity requirements, higher data rates, and additional bandwidth that exceeds 1 GBit/s for mobile access are required to reach this arrangement [1]. It will be extremely challenging to achieve those aggressive 5G performance metrics all at once, and thus, the 5G revolution is expected to occur in stages [2]. In contrast, base stations capable of dealing with multi-band signals have gained widespread popularity, which leads to new research

trends in the linearlization of power amplifiers (PAs) [3–5]. Additionally, bandwidth requirements in conventional radio frequency (RF) paths for multi-band transmission systems involve high nonlinearities and memory effects added by the device under test [4]. In recent years, the cutting-edge development to provide innovative integrated circuits (ICs) integrated along with foundry/technology selection through complementary metal-oxide semiconductor CMOS, gallium arsenide (GaAs), and GaN PA designs have demonstrated an improvement in performance, cost reduction, and digital functionality. Therefore, optimal solutions are conducted to guarantee 5G wireless communications systems with antenna array design integration such as those for massive multiple input/multiple output (MIMO) realizations [2]. It is well-known that the radio frequency power amplifier (RF-PA) performance can often dominate the overall transmitter (TX) behavior, such as the power-added efficiency (PAE), which impacts the power and heat dissipation for the TX chain. The focal point of the RF-PAs is to boost the signal level in order to reach in the receiver chain a signal with suitable power levels to allow for the detection and demodulation process. However, it is the device in the transmission chain that adds most of the nonlinearities and short–term memory effects, which generate the undesirable effect of spectral regrowth and second- and third-order intermodulation (IMD3) products. As it is well known, single-band RF-PAs generate intermodulation, which increases the interference to neighboring channels and the transmission quality [5]. Therefore, special techniques and proposals for corrections during a wireless transmission are required in order to avoid penalties by the telecommunications regulatory entities in each country. To achieve this objective, linearization adaptive techniques and schemes consider the electrical factor of the chain to properly correct short-term memory effects [6].

Models derived from the Volterra Series estimation have been proven to have appropriate accuracy in modeling stages for nonlinear systems. Alternative techniques such as Wiener, Hammerstein, and serial or parallel configurations consider the nonlinear (NL) and linear time invariant (LTI) stages as a suitable strategy in the modeling of nonlinear devices [7,8]. Additionally, memory polynomial model (MPM) and the generalized memory polynomial (GMP) model are related works that make important approximations in this area. A novel output generalized memory polynomial (OGMP) behavioral model based on the previous output signal for linearization of PA was proposed in [9]. Conventional MP or GMP models use polynomials of the previous input signal to characterize memory effects [8]. The behavioral models for PAs devices driven with single and multiband signals have been designed to mimic the nonlinear regimes and to provide a useful approximation of its nonlinear response under several stimuli. Additionally, it is proposed to model a digital predistorter generalized parallel two-box (GPTB) based on hybrid memory polynomial since it consists of a hybrid memory polynomial and a memoryless nonlinear function [10].

Regarding this nonlinear aspect, special efforts are needed to investigate the effects of nonlinear distortion produced by RF-PAs, which include amplitude distortion, phase distortion [11], and the spread of signal constellations based on the quasi-memoryless Saleh model when passed through the additive white gaussian noise (AWGN) channel. This model was chosen because it involves the amplitude-to-amplitude (AM/AM) and amplitude-to-phase (AM/PM) conversion curves [12]. Additionally, the need to establish methodologies for transmitting and modeling data primarily with digital multiplexing such as long-term evolution (LTE) and wideband code division multiple access (WCDMA) are of current interest [13], even implementing a two-chain amplifying structure in parallel for the 2.5 GHz band [14]. However, some linearization techniques for RF-PAs have low performance and limited adaptability regaring data variations due to short-term memory effects [15]. The 1-dB compression point (P1dB) is one of the main electrical characteristics that represents the change between the linear and compression sections of any RF-PA device. The P1dB makes a comparative performance in the linear region and the saturation zone of the nonlinear device [16,17]. In this work, the weighted memory polynomial model (W-MPM) is proposed as an adaptive technique that takes into account the electrical variation of a nonlinear device under test corrected by a indirect learning approach (ILA)

linearization scheme [18]. The variations in time of a device as RF-PA are generated by the impedances and parasitic capacitances within the device; unfortunately, during the amplification process, the devices are brought to the saturation zone, and it is precisely in this region where it behaves as highly nonlinear. A device in this region produces not only in-band distortion but also spectral regrowth. The amplification process after the P1dB generates a low efficiency of the used power; this condition is increased in high frequencies. The state-of-the-art reports different linearization methods that can be developed in the simulation stage prior to implementation; one of them is related to fixed data, where the best correlation data is sought in the output of the system, techniques such as look-up table (LUT) based ones [19]. Other methods based on the data variation are indirect learning approach (ILA) or direct learning approach (DLA). In the ILA stage, a post-distortion is used, where the device's output measurements are taken as the post-distortion input and the input measurements are taken as the post-distortion output. Then, the post-distorter is estimated based on the least square error (LSE) method, and the result is placed before the RF-PA. Unlike DLA, it is required to define an RF-PA model then, and finally, the total linearization is just the inverse result of this model [18,20,21]. Some linearization works related to compensation of nonlinear systems use a compensation model to update the coefficients of a system, such as nonlinear applications of the global system for mobile communication (GSM) [22]; in addition, models derived from Volterra Series such as MPM to compensate for the characteristics of the short-wave PA are presented, where IMD3 to overcome the nonlinear behavior are reduced [23]. Additionally, a two-step approach was developed for DLA as linearization technique based on convolutional neural network [24]. It should be noted that, in the works developed, they do not use an electrical parameter that can vary to adjust the modeling stage or to compensate the system. Therefore, to address the aggressive multi-carrier LTE-A and WCDMA modulation schemes, it is still an important challenge to fulfill the power, efficiency, and linearity requirements in new architectures of RF-PAs [25]. The main aim of this paper is to offer an adaptive alternative to correct the undesirable effect of spectral regrowth, based on modeling and linearization stages, where the P1dB of a nonlinear device caused by the memory effects in short time is considered, the obtained results are tested into a development board, the error vector magnitude (EVM) is evaluated, and the developed adaptive linearization scheme is able to adapt to the nonlinear behavior under analysis. The novelty presented in this work is the integration of an adaptive modeling stage for RF-PA linearization for applications in the 2.5 GHz band for highly nonlinear devices related to telecommunications and for worldwide interoperability for microwave transmissions access (WiMax) and long-term evolution (LTE), where the modeling stage considers the P1dB as a metric for updating coefficients.

The remainder of this paper is organized as follows. In Section 2, the modeling description involving the MPM, W-MPM, and ILA linearization methods are depicted. Section 3 describes the experimental setup used to perform the measurements of the RF-PA under test in addition to the simulation system implementation based on the DSP builder tool prior to implementing it on the FPGA development board. Section 4 shows the obtained results and precision of the highly nonlinear device under test, the modeling accuracy, and the linearization result. Finally, in Section 5, a discussion and the main conclusions obtained are presented.

## 2. Dynamic Modeling Stage

This section describes the traditional MPM and the W-MPM used as the adaptive modeling method that considers the P1dB as the main electrical characteristic. This represents the change between the linear and saturation regions and shows adequate adaptability to the electric factor and data variation during the coefficient estimation process. Additionally, the system is implemented in the Intel® Altera FPGA Cyclone V board through the DSP builder design tool using the 14-bit resolution high speed mezzanine card (HSMC) acquisition board.

### 2.1. Memory Polynomial Model (MPM)

The state-of-the-art reports MPM as an accurate modeling technique with low computational cost related to coefficients calculation and hardware implementation for RF-PA modeling without taking into account the amplification process. Wide RF-PA data variation due to memory effects or changes in the P1dB are an interesting evaluation of the development of an adaptive alternative [26]. The MPM consists of delay stages that involve the diagonal terms of the general Volterra Series; the MPM can be expressed by

$$y(n) = \sum_{k=1}^{K} \sum_{m=0}^{M} a_{k,m} x(n-m) |x(n-m)|^k, \tag{1}$$

where the complex baseband signals are expressed by the $x(n)$ and $y(n)$ input–output relationship. $a_{k,m}$ is the complex polynomial coefficients, $M$ is the memory depth, and $K$ is the polynomial order. In this paper, it is shown that the W-MPM as an adaptive methodology is effective in taking into account electrical variations for modeling highly nonlinear RF-PAs. This improves the accuracy of reported MPM as truncation of the Volterra Series. The benefits of the W-MPM are related to the exhibit nonlinearities of the PA exhibited at low input to voltage levels with strong memory effects in the short term and at a high input voltage level with mild memory effects [27].

### 2.2. Weighted MPM as Dynamical Modeling and Linearization Stages

For high bandwidth applications requiring an additional linearization process, it is necessary to develop accurate models and much better if this modeling stage follows an adaptive methodology due to the nonlinearities added to the wireless transmission. To better address these nonlinearities, the W-MPM incorporates a novel weight function with a tradeoff between the static and dynamic distortions depending on the input value. This is an interesting contribution for the amplification process change due to normal use of the device. These weight functions are involved in the calculation of the coefficients in order to detect when the short-term memory occurs [27]. The W-MPM can be expressed by

$$
\begin{aligned}
y_{W-MPM}(n) = {} & \sum_{k=1}^{K_S} a_k W_S(|x(n)|) x(n) |x(n)|^k \\
& + \sum_{k=1}^{K_D} \sum_{m=0}^{M} \beta_{k,m} W_D(|x(n-m)|) x(n-m) |x(n-m)|^k
\end{aligned}
\tag{2}
$$

where $a_k$ is the coefficient of the static part of the model, $\beta_{k,m}$ is the model coefficients for the dynamic behavior, $K_S$ is the maximum nonlinear order of the static side of the model, and $K_D$ accounts for the memory effects added to the device under test. In this case, $W_S(|x(n)|)$ is related to the static behavior and $W_D(|x(n-m)|)$ represents the weighted functions involved in the dynamics of the model, $k$ represents the incremental level of nonlinearity order, and $m$ represents the short-term memory depth. In both cases, P1dB is the crucial parameter of the calculation. It must be noted that the P1dB is the main electrical factor that affects the coefficient update. Then, $y_{W-MPM}$ is the complex baseband output signal. Therefore, the weighting function for the dynamic part of the model is formulated by

$$W_D(|x(n-m)|) = \frac{1}{2} \left\{ \tanh \left[ G(k,m) \cdot \left( 1 - \frac{|x(n-m)|}{|x|_{th}} \right) \right] + 1 \right\}. \tag{3}$$

where $|x|_{th}$ is the threshold value given by

$$|x|_{th} = x_{th_n} \cdot |x|_{\max}. \tag{4}$$

where $|x|_{th_n}$ is directly related to the P1dB compression point as a criterion for modeling for each RF-PA. On the other hand, $|x|_{max}$ is the maximum voltage of the input value. $x_{th_n}$ is the coefficient related to the P1dB established by the input voltage value in correlation with the amplification result. The $G(k, m)$ function is given by

$$G(k, m) = \frac{1}{k \cdot m^2}. \tag{5}$$

Similarly, the weight function of the static model is described by Equation (6), where the P1dB is the main electrical factor:

$$W_S(|x(n)|) = \frac{1}{2} \left\{ \tanh \left[ -k \cdot \left( 1 - \frac{|x(n-m)|}{|x|_{th}} \right) \right] + 1 \right\}. \tag{6}$$

After having developed the mathematical modeling of the RF-PA according to the W-MPM from the input–output data, the normalized mean square error (NMSE) was established as a metric to estimate the general deviations between the predicted and measured values. The calculation methods and functions used were programmed in Matlab and include the three variants of coefficient extraction for modeling (all, even, and odd). The results that we show in this paper apply to the use of only even coefficients in the model due to a better NMSE value obtained for the cases at hand. The basic principle of the linearization method is to place the nonlinear correction in front of the RF-PA behavioral modeling stage so that the combined data of the nonlinear and linear time-invariant functions produce a linear gain. In this work, the ILA linearization algorithm is proposed as an alternative with a proper accuracy joined with the dynamical stage where the electrical factor as P1dB can be considered during the coefficient extraction. The output of the ILA stage is expressed as

$$z_p(n) = \sum_{m=0}^{M} \sum_{l=0}^{L} b_{ml} \phi_{ml}[x(n)], \tag{7}$$

where $z_p(n)$ represents the relation of $y(n)/G_0$, $G_0$ is the maximum gain obtained from the AM/AM conversion curve, $\phi_{ml}[x(n)] = x(n-m)|x(n-m)|^{2K-1}$, $b_{ml}$ is the complex coefficient of the linearization stage, $M$ is the memory depth, and $K$ is the nonlinearity order.

## 3. Experimental System-Level Setup

In this section, an experimental setup is presented to conduct characterization of the medium high PA. Similarly, the data-driven procedure is used to perform the modeling and the experimental validation. In this section, the measurement setup established based on a FPGA development board, local oscillator, and power sensors is also described, in this case, introducing a signal sweep for the device under test.

*Measurement Procedure*

In accordance with the measurements obtained, a setup with a realistic operation of a base station is realized as shown in Figure 1. At the transmitter/receiver, the signals are first frequency shifted by a digital intermediate frequency (IF) set at 76.8 MHz using dual channel 16-bit digital-to-analog converters (DACs) with a synchronization clock set at 307 MHz. The IF-to-RF up-conversion path uses an IQ modulator driven by a local oscillator (LO) that is used at 2 GHz to be mixed with the baseband signal. The main program structure is created through MATLAB that is connected to the FPGA and can download/upload data frame from/to the synchronous instantiated ROM memory on the FPGA. The signals are generated in MATLAB, stored to RAM instantiated memories, and processed by FPGA (Arria V GX-360KLE) on the board. The FPGA controls all the signal sequences, feeds the DAC board (TI transmitter TSW30H84) with baseband signals, and acquires the digital samples from the analog-to-digital converter (ADC) board (TI

feedback/receiver TSW1266 with a dual-channel ADC ADS5402). In the DAC board, two 32-bit numerically controlled oscillators (NCOs) are configured to shift the two transmitted signal to IF frequencies $\omega_{IF,Tx_1}$ and $\omega_{IF,Tx_2}$, respectively, which are then modulated by a selected radio frequency carrier $\omega_{RF,C}$.

At the mixing process, the signal is up-sampled with a factor 4 filter interpolation using a sampling rate of 1.2 GHz in the up-conversion. Note that the local oscillator power of $-11$ dB also contributes to the output power, a calibration procedure is given in order to set an adequate input level at the $P_{in}$, and then the resulting signal is amplified through a predriver (Mini-Circuits ZRL-2300+). A 30 dB attenuator is placed at the PA output simplification to perform PA characterization and measurements of a medium-high PA (Mini-Circuits ZHL-42W+). Given the FPGA design, it is identified that low-voltage differential signaling (LVDS) serializer/deserializer (SERDES) circuitry introduces the most prominent latency across the channel Tx/Rx. A time alignment is jointly performed from a cross-correlation simultaneously from the real-time at the Tx/Rx data, in concordance. Moreover, during the experiments to provide fair comparisons with the DPD methods, a wide-band receiver is used to acquire the PA output at a sampling rate of 614 Msps, which is twice the transmission bandwidth: 307 MHz.

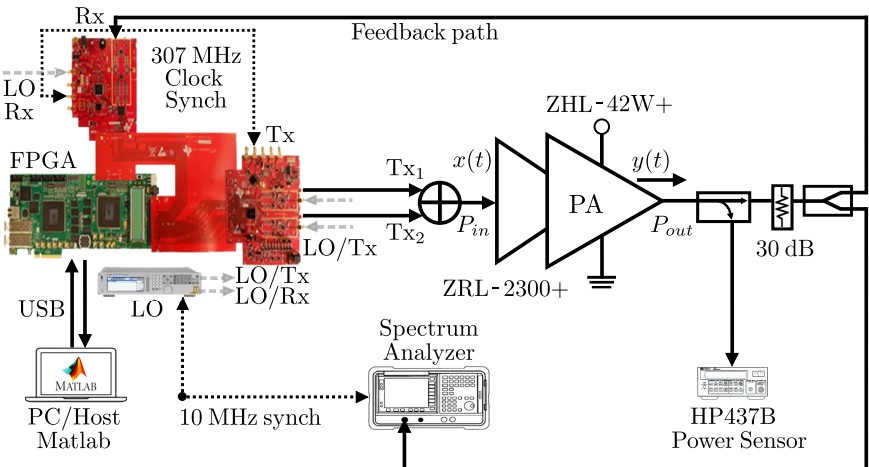

**Figure 1.** Experimental measurement testbed setup based on ARRIA V Board for a radio frequency power amplifier (RF-PA).

Table 1 shows the main electrical characteristics of the PA NXP Semiconductors 10 W, at 2.34 GHz, and Table 2 show that of the device ZHL-42W+ at 2000 MHz 32.24 dBm under test; they contain the biasing and main setting established during the performed device measurements.

**Table 1.** Features of the NXP 10 W PA.

| PA NXP 10 W @ 2.34 GHz | |
|---|---|
| **Parameter** | **Values** |
| Gain | 12.26 dB @ 2.34 GHz |
| PA input Power | 23.84 dBm |
| PA output Power | 36.10 dBm |
| Device biasing | $V_{DS}$ = 50 V, $I_{DS}$ = 54 mA |

**Table 2.** Features of the ZHL-42W+ PA at 32.24 dBm.

| PA ZHL-42W+ @ 2.00 GHz | |
|---|---|
| **Parameter** | **Values** |
| Gain | 34 dB @ 2.00 GHz |
| Maximum power | 1.25 W |
| DC Supply | 15 V, 1 A |
| Bandwidth | 10–4200 MHz |

## 4. Modeling and Linearization Stage Setup

The system level model is illustrated in Figure 2 with the designed modeling and system-level stages in Simulink to develop the modeling and linearization for the chain of the PA device under test. The overall structure is flexible up to a nonlinear order of 11 and memory 6. Figure 2a depicts the output string of the DSP builder tool, where the data are controlled with a resolution of 10 bits in the bus address and the amplitude is sampled with the maximum resolution of 14-bit. In this case, the output signals are divided into 16,384 parts. In this scheme, the coefficient calculation is obtained through code based on the LSE method and can be updated taking into account the electrical factor P1dB that corresponds to the voltage when the linear gain corresponds to a 79.43% decrease.

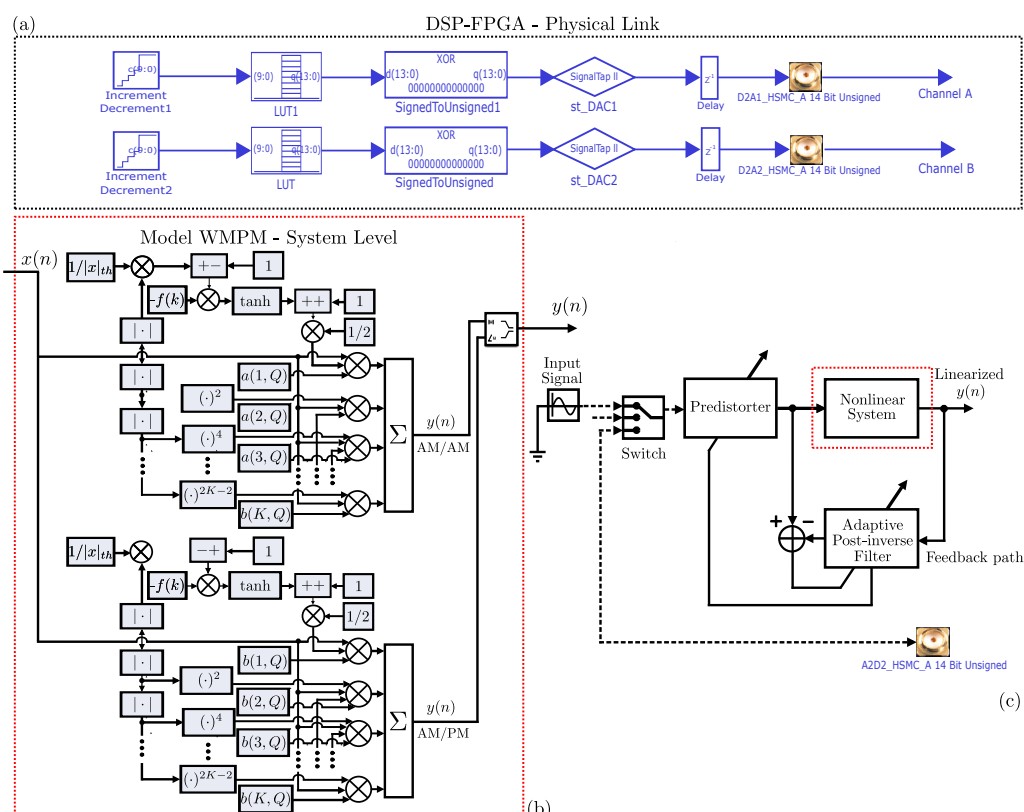

**Figure 2.** Indirect learning approach (ILA) algorithm block diagram: (**a**) DSP-FPGA physical link for data conditioning, (**b**) adaptive weighted memory polynomial model (W-MPM) modeling stage, and (**c**) proposed ILA system control-based scheme.

### *Indirect Learning Approach for the Proposal Modeling Stage*

Behavioral modeling of the medium-high RF-PA was developed through the MPM expressed by Equation (1) and the W-MPM expressed by Equation (2). The coefficient extraction process was done using the LSE method. The development structure depicted in Figure 2b corresponds to the W-MPM proposed as the adaptive modeling taking into account the electrical factor of P1dB; a similar structure is performed for the MPM. This

schematic is used to model the NXP 10 W at 2.34 GHz and ZHL-42W+ at 2000 MHz 32.24 dBm RF-PAs. The experiments conducted are specified below:

(i)    An algorithm is performed for the offline training of the model (PA) and the predistorter (PD) block using the MPM and W-MPM approaches (running on MATLAB-PC with DSP blockset system environment) as defined in the overall block diagram in Figure 2.

(ii)    At every iteration, the model searches for a weighting subset of parameters to contribute in the minimization the LSE and NMSE.

(iii)    The developed chain through DSP Builder tool allows us to transfer the input signal compared with the amplification process by the DAC of the FPGA development board Cyclone V.

(iv)    Both signals are sampled using 10-bit resolution related to the address bus capability; the magnitude signals are sampled for the maximum resolution of the HSMC card with 14 bits.

We evaluate in this work both conversion curves that present a notable presence of highly nonlinear behavior. The achieved error was $-32.72$ dB for the MPM and $-38.03$ dB in NMSE for the W-MPM. Additionally, it was performed a modeling stage obtaining $-44.34$ dB NMSE based on the MPM and $-44.90$ dB NMSE for the device ZHL-42W+ at 2000 MHz, and it should be noted that the devices have a very high nonlinear behavior and strong data dispersion. Both models contain more than 65 K data samples for each conversion curve. The W-MPM showed a better accuracy compared with the traditional MPM, and it can even be updated if the conversion curves change with time due to its intrinsic adaptability based on the P1dB detecting the compression region in the amplification stage. During hardware implementation, the total resources used for each model is compared. In this case, it obtained a slightly reduced amount of the available resources in the FPGA board Cyclone V during the W-MPM compared with the MPM, as it can be seen in Table 3. The W-MPM had a reduction in the adaptive logic module (ALM) using 340 compared with the required 378 used by the MPM.

**Table 3.** Hardware resources occupation in Cyclone V 5CEFAF31I7 FPGA device with comparison for the W-MPM and MPM models.

| Description | W-MPM Resource Utilization | MPM Resource Utilization |
|---|---|---|
| Logic Utilization (ALMs) | 340/56,480 (<1%) | 378/56,480 (<1%) |
| Total Registers | 771 | 751 |
| Total Pins I/O | 40/480 (8%) | 40/480 (8%) |
| Total block memory bits | 43,008/7,024,0640 (<1) | 43,008/7,024,0640 (<1) |
| Total PLLs | 1/7 (14%) | 1/7 (14%) |

Table 4 summarize the Halstead metrics based on the two developed modeling stages W-MPM and MPM. The following estimation was calculated based on the developed code for the extraction coefficient method and the implementation chain. As it can be seen, the complex operations associated with the program length indicate the difficulty and effort during both processes, which is considerably bigger for the MPM compared with the W-MPM. However, both cases were resolved in a considerable reduced time for this type of implementation, where the model has a big data dispersion between short value intervals. A Halstead complexity analysis was done for both modeling stages where the W-MPM requires less offline training before implementation compared with the MPM in this study case, which required 11.85 s compared with 13.33 s.

**Table 4.** Halstead complexity analysis for W-MPM and MPM.

| Complexity Metric | W-MPM | MPM |
|---|---|---|
| Number of distinct operators | 7 | 7 |
| Number of distinct operands | 7 | 7 |
| Total number of operators | 9 | 11 |
| Total number of operands | 7 | 7 |
| Program vocabulary | 14 | 14 |
| Program length: | 16 | 18 |
| Calculated estimated program length | 39.30 | 39.30 |
| Volume | 60.92 | 68.53 |
| Difficulty | 3.50 | 3.50 |
| Effort | 213.21 | 239.86 |
| Time required to program (s) | 11.85 | 13.33 |
| Number of delivered bugs | 0.07 | 0.08 |

Figures 3 and 4 show the linearization performance based on MPM and W-MPM, respectively; in both cases, the amplitude and phase distortion were properly linearized. The results obtained from the adaptive W-MPM algorithm minimized the required resources and coefficients compared with the MPM. These models and the learning code were implemented in FPGA development board Cyclone V, and it was achieved by reducing the amount of coefficients and by improving the accuracy compared with reported works in the literature. The accuracy obtained was $-32.72$ dB NMSE using the MPM compared with $-38.03$ dB NMSE for adaptive W-MPM.

Figures 5 and 6 show the developed models that replicate effectively the highly nonlinear device under analysis. For both devices, it was estimated the EVM after the linearization algorithm and the distortion between the receiver and transmitter chain were reduced. In this case, it was improved by 13.07% and by 15.48% for the EVM for the NXP 10 W at 2.34 GHz using the static MPM and dynamic W-MPM, respectively. For the ZHL-42W+ device operating at 2000 MHz with an output power of around 32.24 dBm, it was improved by 24.1% and 24.22%, while the EVM is in the same conditions before and after the proposed linearization algorithm.

Table 5 summarizes the performance comparison with a study of the involved works in the state-of-the-art comparing mainly the RF-PA linearity order required to properly address the behavior under a complexity analysis. Using the aforementioned simplification the analysis assumes the required coefficient number of the modeling stages and yields a feasible accuracy. Additionally, it is depicted as the technology of the RF-PA under experimentation. In Figure 7 is depicted the development board Cyclone V and acquisition card. Since the contribution of this paper is to offer an adaptive linearization scheme, in this case, the baseband components with the up conversion cards are not required to frequency shift from IF to baseband or even to use physical attenuators to attenuate the signal power levels.

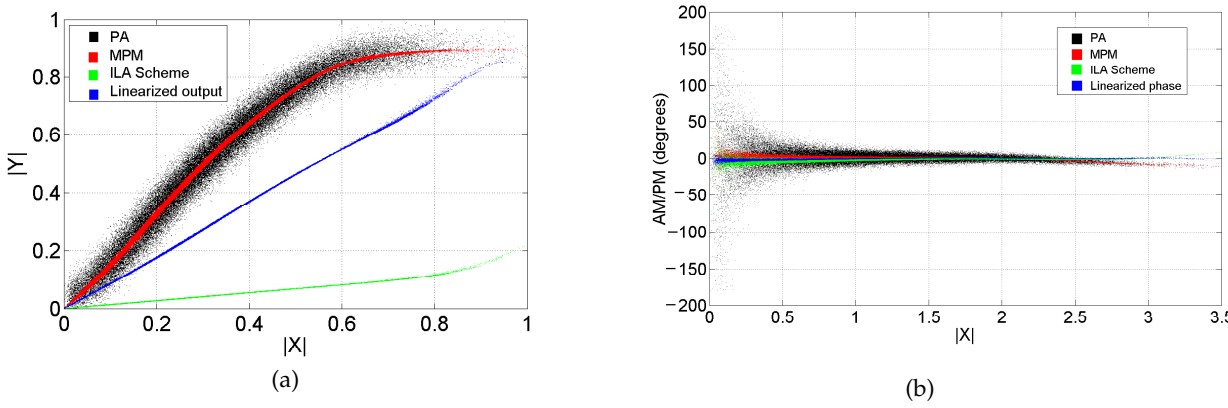

**Figure 3.** Linearization based on ILA based on MPM for the device NXP 10 W at 2.34 GHz. (**a**) AM/AM modeling and linearization stage and (**b**) AM/PM modeling and linearization stage.

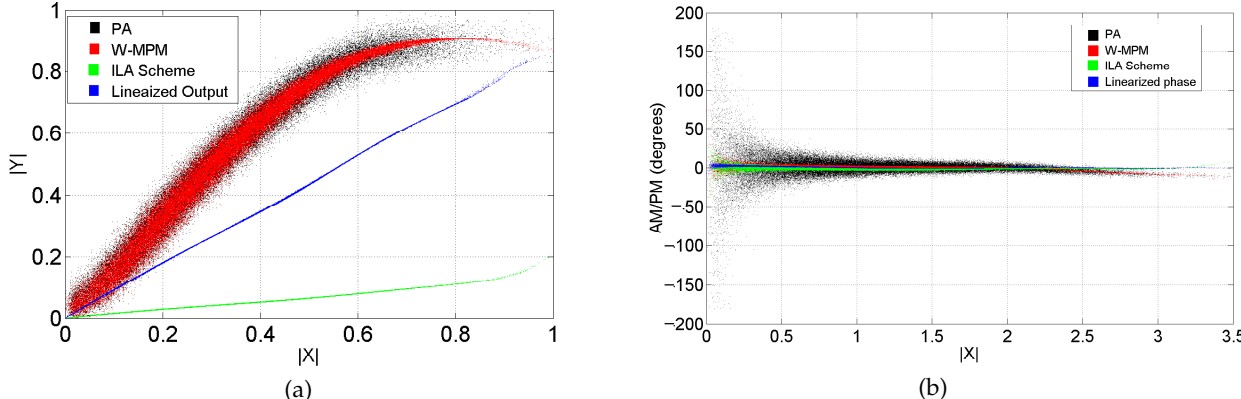

**Figure 4.** Linearization based on ILA based on W-MPM for the device NXP 10 W at 2.34 GHz. (**a**) AM/AM modeling and linearization stage and (**b**) AM/PM modeling and linearization stage.

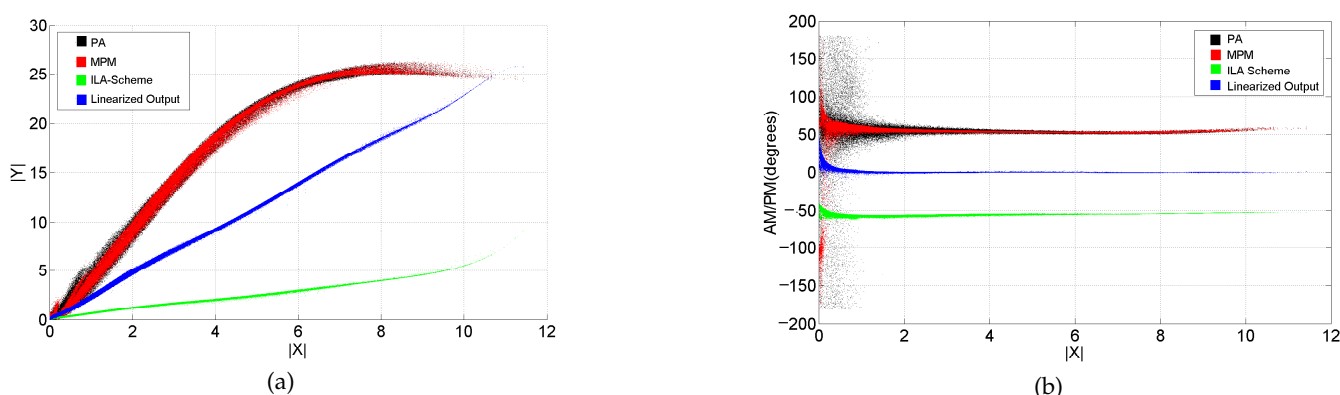

**Figure 5.** Linearization based on ILA based on MPM for the device ZHL-42W+ at 2000 MHz 32.24 dBm RF-PA. (**a**) AM/AM modeling and linearization stage and (**b**) AM/PM modeling and linearization stage.

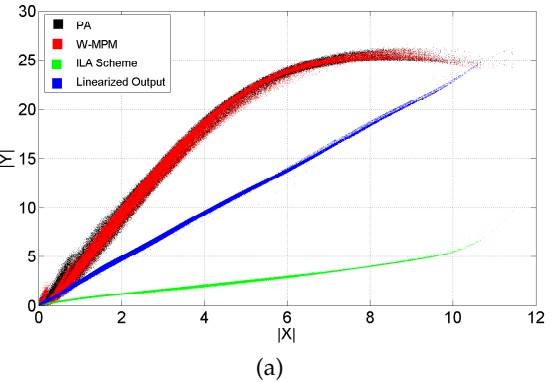
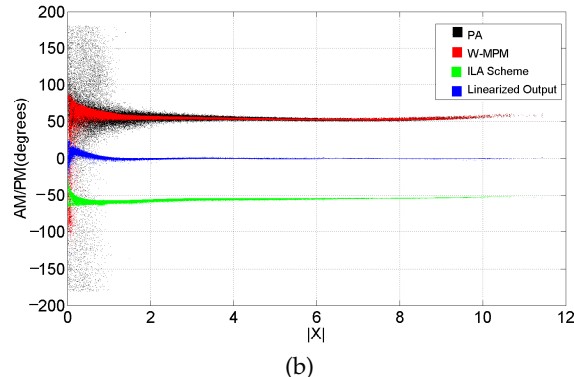

(a)          (b)

**Figure 6.** Linearization based on ILA based on W-MPM for the device ZHL-42W+ at 2000 MHz 32.24 dBm RF-PA. (**a**) AM/AM modeling and linearization stage and (**b**) AM/PM modeling and linearization stage.

Table 5 summarizes a comparison of the proposed work as a dynamical modeling stage compared with other related works. In addition to this, a model derived from the general Volterra Series such as the MPM is described; the W-MPM is proposed as an adaptive modeling technique in a comparison process where the fundamental basis of the dynamic weighting selection proposal is considered as a modeling stage for estimation of the distortion curves. In this case, the P1dB compression point of the device is identified by programming and represents the point at which there is a gain level of 79.43% in relation to the expected results based on its linear area of the device. The dynamic factor is automatically calculated and established as the $X_{th}$ value. Furthermore, the model is analyzed in two stages, dynamic and static parts, and is able to update the coefficient in case of wide variation in the amplification process due to the memory effects in short time.

An adaptive and static modeling technique is used to estimate the highly nonlinear behavior of the PA NXP 10 W at 2.34 GHz. In this case, the precise modeling stage is crucial, which can be used for a linearization technique; in this paper, we have shown the ILA technique that was compatible with modeling stage based on MPM and W-MPM.

The adaptive modeling stage reaches for the NXP semiconductor of 10 W GaN HEMT RF-PA an accuracy of −32.72 dB NMSE using the MPM compared with −38.03 dB for the dynamical modeling stage based on the extraction algorithm. Additionally, it compared the highly nonlinear behavior of the device ZHL-42W+ at 2000 MHz 32.24 dBm, where an NMSE accuracy of the modeling stage of −44.34 dB was reached for the MPM, while for the W-MPM adaptive algorithm, it is around −44.90 dB. In addition to this, the used hardware resources on the Cyclone V development card show a usage of 340 adaptive logic modules for the W-MPM compared with the 378 used resources for the MPM (see Table 3). The transmission chain involves two channels designed for handling sampled data with resolution $2^n$ of the input signal with respect to the outputs. It is shown a voltage offset block stage for adjustment linked to the DACs. In the simulation stage, the coefficients of the modeling stages are calculated through the LSE method for the MPM and W-MPM, The even terms are used in both conversion curves estimation since they give the best precision based on NMSE error. An attenuator was designed for the voltage unbalance of the input signal and the maximum peak voltage of the amplified signal. The most important contribution of this work lies in the proposal of a W-MPM model and ILA scheme for highly nonlinear models and data dispersion in the same way a whole dynamic weighting selection system is developed where the change in amplitude can be monitored.

The W-MPM yields a considerably lower number of coefficients compared to the MPM; the related P1dB parameter that involves 7 coefficients plus one per nonlinear stage in total generates 42 coefficients for the device under test. In the static part, since it does not consider memory, the design is implemented with a W-MPM with nonlinearity of 13 and memory effect of zero, which gives a total of 55 coefficients. In relation to the MPM, to obtain relatively the same result based on the metric NMSE, a total of 67 coefficients are

involved and a system of nonlinearity order 11 and high memory with level 6 is required. The EVM reduction is estimated for the developed static and dynamic modeling stages; both linearization process are compared. As we can see, the achieved EVM reduction is considerably better after the linearization stage. The improved EVM is shown in Table 6.

**Table 5.** Comparison of dynamic W-MPM for RF-PA modeling versus related work.

| Model Stage | RF-PA Linearity | Technology | Nonlinearity and Memory Effects | Coefficients Number | Accuracy NMSE (dB) |
|---|---|---|---|---|---|
| **Proposed work, Device: PA NXP 10 W @ 2.34 GHz** | | | | | |
| W-MPM model | Nonlinear | GaN HEMT | High order | 55 | −38.03 and −44.9028 |
| MPM model | Nonlinear | GaN HEMT | High order | 67 | −32.72 and −44.349 |
| **Proposed work, Device: ZHL-42W+ @2000 MHz 32.24 dBm** | | | | | |
| W-MPM model | Nonlinear | CMOS+LVDS | High order | 55 | −27.8946 |
| MPM model | Nonlinear | CMOS+LVDS | High order | 67 | −24.8707 |
| **Related works** | | | | | |
| Hammerstein [†], [28] | Nonlinear | GaN | High order | N/A | −33.55 |
| Hammerstein [‡], [28] | Nonlinear | GaN | High order | N/A | −35.72 |
| MP, [29] | Nonlinear | GaN Doherty | High order | N/A | −32.2 |
| EMP, [29] | Nonlinear | GaN Doherty | High order | N/A | −24.9 |
| SVR, [30] | Nonlinear | LDMOS | High order | 256 | −36.5 |
| DVR, [31] | Nonlinear | GaN Doherty | High order | 99 | −31 |

[†] Conventional Hammerstein model. [‡] Modified Hammerstein model.

**Table 6.** EVM improvement based on the ILA algorithm for the MPM and W-MPM models.

| Device NXP 10 W @ 2.34 GHz | Estimated EVM | EVM with ILA |
|---|---|---|
| MPM | 15.624% | 2.547% |
| W-MPM | 16.54% | 1.06% |
| **Device ZHL-42W+ @ 2000 MHz 32.24 dBm** | **Estimated EVM** | **EVM with ILA** |
| MPM | 25.28% | 1.18% |
| W-MPM | 25.18% | 0.96% |

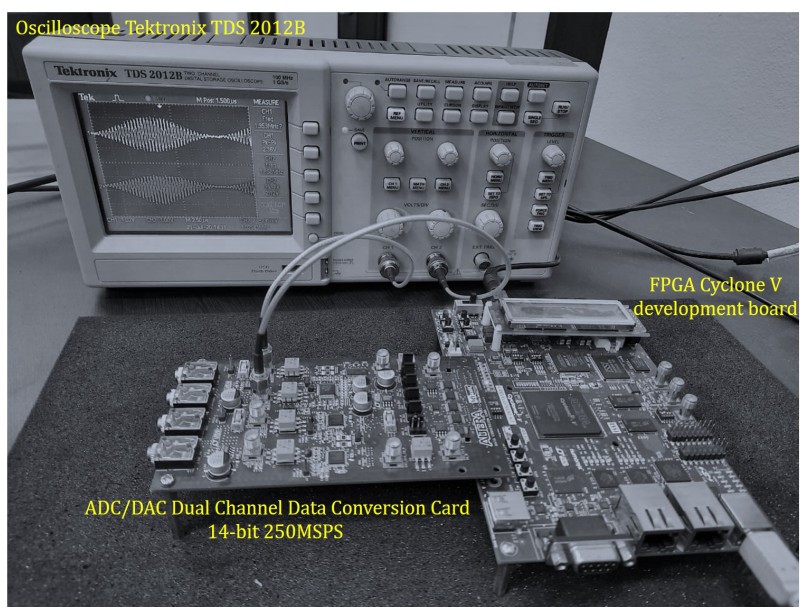

**Figure 7.** Photograph of setup hardware implementation of the adaptive modeling W-MPM stage.

## 5. Conclusions

A dynamic weighting selection system based on two modeling stages, W-MPM and MPM, applied to estimate the phase between signals was developed in this work. In this matter, an effective modeling stage was implemented in hardware to properly address highly nonlinear behavior. In this regard, the W-MPM required 55 and the MPM 67 coefficients to represent the device PA NXP 10 W at 2.34 GHz behavior. The adaptive modeling stage reaches for the NXP Semiconductor of 10 W GaN HEMT RF-PA an accuracy of $-32.72$ dB NMSE using the MPM compared with $-38.03$ dB NMSE for the W-MPM for the 10 W GaN HEMT RF-PA and of $-44.9028$ dB NMSE based on the MPM and $-44.349$ dB NMSE for the device ZHL-42W+ at 2000 MHz, respectively.

The Halstead complexity analysis done establishes that the volume, complexity, and timing convergence to implement both models is 11.85 for the W-MPM and 13.33 s for the MPM, and the achieved time is relatively within the expected range due to the involved iterations in the whole developed design. It is also highlighted that the FPGA implementation required less than one percent of the available logical units; the correction in amplitude and phase based on the ILA scheme was properly performed. The EVM was estimated before and after the linearization algorithm and modeling stages based on MPM and was improved by 13.07% and 15.48% for the NXP 10 W at 2.34 GHz using the static MPM and dynamic W-MPM, respectively. In the case of the ZHL-42W+ at 2000 MHz 32.24 dBm, it was improved by 24.1% and 24.22% for the EVM in the same conditions before and after the proposed linearization algorithm.

**Author Contributions:** Conceptualization, J.A.G.-A., C.V.-R. and J.R.C.-V.; methodology, J.A.G.-A., C.V.-R., E.I.-G., J.C.N.-P. and J.R.C.-V.; software, J.A.G.-A., Y.M.-R., Y.S.-I. and J.R.C.-V.; validation, J.A.G.-A. and J.R.C.-V.; formal analysis, J.A.G.-A., J.C.N.-P. and J.R.C.-V.; investigation, J.A.G.-A., J.R.C.-V. and C.V.-R.; resources, C.V.-R. and J.C.N.-P.; writing—original draft preparation, J.A.G.-A., C.V.-R. and J.R.C.-V.; writing—review and editing, C.V.-R. and E.I.-G.; supervision, C.V.-R.; funding acquisition, C.V.-R. All authors have read and agreed to the published version of the manuscript.

**Funding:** The authors wish to thank TECNM for its support provided through the research projects 7926.20-P and 2021. Also, authors appreciate to Everardo Inzunza-González that collaborate in a sabbatical year authorized by the Autonomous University of Baja California Academic Commission with official letter 21/2020.

**Institutional Review Board Statement:** Not applicable.

**Informed Consent Statement:** Not applicable.

**Data Availability Statement:** Not applicable.

**Acknowledgments:** This work was supported in part by the SEP-CONACyT Research Project under grant 255387 and grant 256237, in part by the School of Engineering and Sciences, and in part by the Telecommunications Research Group at Tecnologico de Monterrey. In addition, the authors would like to express their gratitude to the IPN for its support by the project "SIP-20210345. The authors would like to thank PRODEP for supporting the new generations and for innovating the application of knowledge.

**Conflicts of Interest:** The authors declare no conflict of interest.

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
