# Peer review of "A Weighted Linearization Method for Highly RF-PA Nonlinear Behavior Based on the Compression Region Identification"

_applsci, doi:10.3390/app11072942_

Round 1

Reviewer 1 Report

It is interesting paper presenting linearization method for RF PA. I have few minor comments. 1 .Figures are not fully shown in the paper. It should be updated. 2. There are some typos in line 261, 269, and more. Please carefully update all the errors. 3. English should be improved for the better readability. 4. Are the Figure 3 ~ 6 measurement data? Please clearly state it.

Author Response

We thank the reviewer for your careful review and time on our manuscript. Please find attached a file with a point-by-point answered to all your points.

Reviewer 2 Report

The authors presented the Weighted Memory Polynomial W-MPM algorithm and validated its implementation and measured its performance. 

Please specify the signal used to plot AMAM and AMPM graphs: the signal bandwidth, the PAPR... 

Can the author comment on using Direct Learning with this model? Can this model be extended to GMP and make it W-GMP? 

The overall quality of work is good. 

Minor spelling errors and typos. Some formatting, text alignment is needed to fit the graphs in the page

Add figure numbers in lines 261 269 282 332 (??)

Author Response

(The authors gave the same response as above.)
